PREPARED FOR SUBMISSION TO JHEP

# Spectral weight in holography with momentum relaxation

**Victoria L. Martin**[1]

[1]*Department of Physics, Arizona State University, Tempe, AZ 85287, USA*

*E-mail:* [victoria.martin.2@asu.edu](mailto:victoria.martin.2@asu.edu)

ABSTRACT: Holographic low-energy spectral weight at zero temperature and finite momenta indicates the presence of a strongly coupled remnant of Pauli exclusion. Building upon previous work, we study the spectral weight of a bottom-up holographic superfluid model with spontaneously broken translational symmetry. We determine the effect of this symmetry breaking on the previously known attributes of the holographic superconductor spectral weight: 1) an instability at finite momenta and 2) the presence of nested Fermi surfaces (sometimes called Fermi shells). We find that the symmetry breaking seems to strengthen the former and suppress the latter, in a way that we describe.

# 1 Introduction

The AdS/CFT correspondence [1] provides an avenue to indirectly study aspects of strongly interacting quantum field theories. A system of particular interest is the so-called non-Fermi liquid phase describing the normal state of high-temperature cuprate superconductors [2]. Some properties of non-Fermi liquids have already been realized holographically, notably the famous linear scaling of resistivity and specific heat with temperature [3]. Another attribute endemic to non-Fermi liquids is that they form Fermi surfaces in momentum space at low temperatures, which can be seen for example by applying an external magnetic field that destroys the superconducting dome [2].

A principal diagnostic for the presence of a Fermi surface is the low-energy spectral weight (see, for example, [4])

$$\sigma(k) = \lim_{\omega \to 0} \frac{\text{Im}G_{\mathcal{O}\mathcal{O}}^R(\omega, k)}{\omega}. \tag{1.1}$$

Here the operator $\mathcal{O}$ can be, for example, the charge density $J^t$ or current $J^x$, but for our purposes we will be interested in $\mathcal{O} = J_\parallel$ and $\mathcal{O} = J_\perp$, corresponding to the transverse and longitudinal channels of the perturbed bulk fields (to be introduced in subsequent sections). There are two different senses in which (1.1) can indicate the presence of a Fermi surface. First, experimental techniques such as angle-resolved photoemission spectroscopy (ARPES) detect a Fermi surface via a pole in the retarded Green's function of (1.1) at $k = k_F$, when $\mathcal{O} = \psi$ [5]. The Green's function in (1.1) is the UV Green's function, and so holographically we need to consider the full bulk geometry to gain access to this pole.

Second, at low energies we have [4, 6]

$$\text{Im}G_{\mathcal{O}\mathcal{O}}^R(\omega, k) \propto \text{Im}\mathcal{G}_{\mathcal{O}\mathcal{O}}^R(\omega, k). \tag{1.2}$$

While this expression allows us to directly relate the IR Green's function $\mathcal{G}^R$ to the UV one, we lose all information about a possible pole, which is stored in the proportionality constant of (1.2). However, we can still infer the presence of low energy spectral weight via the spectral decomposition [4]:

$$\text{Im}G_{JJ}^R(\omega, k) = \sum_{m,n} e^{-\beta E_m} \left| \langle n(k^{'})|J(k)|m(k^{''})\rangle \right|^2 \delta(\omega - E_m + E_n). \tag{1.3}$$

The expression (1.3) contains two delta functions, one in energy and one in momentum (resulting from the inner product). Thus we see that the spectral weight directly counts *charged* degrees of freedom (charged due to the presence of $J$) at a given frequency and momentum. In particular, for a field theory at zero temperature the presence of low (zero) energy (frequency) spectral weight at a finite momentum would suggest a remnant of the Pauli exclusion principle, even in the absense of single-particle excitations. Thus we can infer the presence or absense of a Fermi surface by considering IR data alone. A more comprehensive exposition of the two preceding paragraphs is given in the Introduction of [7] and in Appendix B.

The spectral weight has been calculated in IR geometries in several holographic theories. For the Einstein-Maxwell-dilaton (EMD) theory in an IR hyperscaling violating geometry (characterized by dynamical critical exponent $z$ and hyperscaling violating exponent $\theta$), [8, 9] showed that low-energy spectral weight is exponentially suppressed.

However, it was discovered that in the limit $z \to \infty$ with the ratio $\eta = -\theta/z$ held fixed the geometry develops fermionic properties. That is, low-energy spectral weight exists in these so-called semi-local quantum liquid geometries (or $\eta$ geometries for short)[1] for EMD in $d = 4$ [10] and $d > 4$ [7], the holographic superconductor [11, 12], and the holographic superfluid with an additional Chern-Simons term [7]. The calculation of spectral weight in holographic superconductors [12] led to some intriguing results:

1. There exists an instability at finite momentum.

2. There exists nonzero low energy spectral weight at finite momentum.

3. A Fermi shell exists[2].

The interpretation of the first point put forward in [12] is that, within a certain range of parameter space, the semi-local quantum liquid geometry is not the true ground state of this theory. Indeed, some high-temperature superconductors have been seen to exhibit a charge density wave phase that coexists with (or perhaps competes with) the superconducting phase [13]. Thus perhaps the true groundstate of our system is a spatially modulated phase[3].

The second result is quite surprising. In the case of the holographic superconductor, the bulk charge density manifestly forms a condensate, and thus one should expect to find a corresponding vanishing spectral weight at finite momentum in the boundary field theory. However, this is not borne out in the holographic calculation of the retarded Green's function. A clear interpretation of this seemingly paradoxical result is still an open problem. If the bulk charge density is indeed meant to correspond to the boundary charge density in a meaningful way, perhaps there are other unaccounted for bulk degrees of freedom responsible for the nonzero spectral weight.

For the third result, it has been shown more recently that these Fermi shells are more pervasive in holographic bottom-up calculations than was previously supposed [7], at least when considering $\eta$ geometries. Fermi shells are known to appear in top-down constructions, for example in $\mathcal{N} = 4$ supersymmetric Yang-Mills [15] and in ABJM theory [16]. Unlike in bottom-up models, in these top-down constructions the dual field theory is explicitly known, and the Fermi shell is known to result from overlapping Fermi surfaces of two distinct species of fermions.

---

[1]See [6] for a beautiful review of semi-local quantum liquids. We will also define this geometry more fully in the main body of this paper.

[2]The two types of low-energy spectral weight that we will encounter are when $\sigma(k) \neq 0$ for $k < k_*$ (which we call a smeared Fermi surface) and $\sigma(k) \neq 0$ for $k_- < k < k_+$ (which we call a Fermi shell).

[3]A similar conclusion was reached in [14].

In this work, we investigate the extent to which the three phenomena described in the previous paragraphs (the finite $k$ instability, the nonzero low-energy spectral weight, and the presence of a Fermi shell) persist in the presence of explicitly broken translation invariance. We accomplish this by adding massless scalar fields proportional to one of the coordinates (so-called "axion" terms) $\psi_i x_i$ to the bottom-up model of the holographic superconductor

$$S = \int d^4 x \sqrt{-g} \left[ R - \frac{1}{2} \partial \phi^2 - \frac{1}{4} Z(\phi) F^2 - \frac{1}{2} Y(\phi) \sum_i \partial \psi_i^2 - \frac{1}{2} W(\phi) A^2 - V(\phi) \right]. \tag{1.4}$$

Einstein-Maxwell-dilaton-axion (EMDA) theories have been studied previously in the contexts of neutral and charged transport [17, 18] and the study of shear viscosity [19]. In this work, we study a toy model of a theory that exhibits both a spontaneously broken $U(1)$ symmetry (as in the holographic superconductor) and explicitly broken translational symmetry (by adding axion terms), Our motivation for breaking translation invariance in this way is that it provides a toy model for studying the effect analytically, subverting the need to construct more complicated phases, such as spatially modulated phases, numerically. We investigate the issue of anomalous low-energy spectral weight in the presence of a condensate found in [12] by examining the effect of varying condensate charge and axion strength on the size of the Fermi surface, both separately and together. This should be regarded as a sister work to [7].

In Section 2 we review the relevant spectral weight analysis of the holographic superconductor as carried out in [12], and add to that work by addressing the effect of changing the condensate charge $W_0$ on the size of the Fermi surface. In Section 3 we compute the low energy spectral weight in the EMDA theory, and in Section 4 we put it all together and study a holographic superfluid model with explicitly broken translation invariance. We end with a discussion of our results and conclusions in Section 5. In Appendix B we offer a more thorough review of the quantity (1.1) and the sense in which we use it to diagnose Pauli exclusion.

## 2   Holographic Superconductor

The low energy spectral weight of the holographic superconductor in the semi-local quantum liquid geometry was analyzed in [12], and we refer the reader to this resource for a more detailed description. In this section we add to that work by addressing the effect of changing the condensate charge $W_0$ on the size of the Fermi surface, which we define below.

The Lagrangian describing this theory is given by

$$S = \int d^4x \sqrt{-g} \left[ R - \frac{1}{2}\partial\phi^2 - \frac{1}{4}Z(\phi)F^2 - \frac{1}{2}W(\phi)A^2 - V(\phi) \right], \qquad (2.1)$$

Here, and in all of the theories that we will consider, we take the coefficient functions to have the following IR scaling behavior:

$$V(\phi) = V_0 e^{-\delta\phi}, \qquad Z(\phi) = Z_0 e^{\gamma\phi}, \qquad W(\phi) = W_0 e^{\epsilon\phi}. \qquad (2.2)$$

This is to ensure that we have a scaling solution, which is motivated by top-down realizations of holographic superfluids from string theory [20–25]. We consider a one parameter family of background geometries labeled by $\eta$:

$$ds^2 = r^{-\eta} \left( \frac{-dt^2 + dr^2}{r^2} + dx^2 + dy^2 \right). \qquad (2.3)$$

This metric is a special limit of the hyperscaling violating geometries, labeled by dynamical critical exponent $z$ and hyperscaling violating exponent $\theta$ (see for example [26]). The metric (2.3) is obtained from the hyperscaling violating one by taking $z \to \infty$ while holding $\eta \equiv -\theta/z$ fixed. Our background gauge and scalar fields have the following profiles

$$A = A(r)dt, \qquad A(r) = r^{\zeta-1}, \qquad \phi(r) = \kappa \log r \qquad (2.4)$$

where $\zeta$ is a constant, free parameter in the theory, and $\kappa$ is a constant that will be fixed by the background equations of motion. To recover the pure EMD theory (as studied in [10]), one fixes $\zeta = -\eta$ (this is equivalent to setting $W_0 = 0$).

## 2.1 Transverse Channel

All perturbations to the background ansatz take the plane wave form $\delta X = \delta X(r)e^{i(kx-\omega t)}$. The transverse channel is characterized by those perturbations which are odd under the transformation $y \to -y$:

$$\{\delta A_y, \delta g_{ty}, \delta g_{xy}\}. \qquad (2.5)$$

Throughout the paper we work in radial gauge $\delta g_{r\mu} = 0$. Here we restate the result reported in [12], which is the existence of low energy spectral weight below the critical momentum $k_\star$:

$$\sigma(k) = \lim_{\omega \to 0} \frac{\mathrm{Im}G^R_{JJ}(\omega, k)}{\omega} \propto \lim_{\omega \to 0} \omega^{2\nu_- - 1} = \begin{cases} \infty & k < k_\star \\ 0 & k > k_\star \end{cases} \qquad (2.6)$$

where

$$\nu_- = \frac{1}{2}\sqrt{5 + 2\eta + \eta^2 + 4k^2 - 4\sqrt{(1+\eta)^2 + 2(1-\zeta)k^2}} \qquad (2.7)$$

and

$$k_\star^2 = \frac{1}{4}\left(-4\zeta - \eta(\eta+2) + 2\sqrt{2\left(2\zeta^2 + \eta(\zeta+1)(\eta+2)\right)}\right). \tag{2.8}$$

Since $\nu_-$ is real in the allowed parameter space, there is no instability in the transverse channel. We say that $k_\star$ defines the *size* of the Fermi surface, since this is the critical momentum above which the spectral weight vanishes.

It is interesting to recast the analysis of the Fermi surface given by $k_\star$ in terms of the condensate charge $W_0$. This is because, from the original analysis of the holographic superconductor [27], we know that the critical temperature for condensation grows monotonically with the charge of the complex scalar, making it easier to condense at large charge. Thus we might expect the size of the Fermi surface $k_\star$ to decrease as a function of $W_0$. One caveat behind this intuition is that the presence of low energy spectral weight in the holographic superconductor is surprising in its own right, and may somehow be related to other degrees of freedom apart from the condensate. Nevertheless, we will see that the spectral weight in the transverse channel supports this naïve intuition.

The full reduced parameter space found in [12] is:

$$\left(0 < \eta \le \frac{1}{2}\left(\sqrt{5}-1\right) \text{ and } -\eta < \zeta < \frac{\eta^2}{2}\right) \text{ or } \left(\eta > \frac{1}{2}\left(\sqrt{5}-1\right) \text{ and } -\eta < \zeta < \frac{1-\eta}{2}\right) \tag{2.9}$$

To translate this into a parameter space involving $W_0$, we note that the background equations of motion fix $W_0$ to be $W_0 = (\zeta+\eta)(1-\zeta)$. Thus $\zeta$ has two roots:

$$\zeta = \frac{1-\eta \pm \sqrt{(\eta+1)^2 - 4W_0}}{2}. \tag{2.10}$$

The positive root corresponds to $\zeta \to 1$ as $W_0 \to 0$. Since $\zeta = 1$ eliminates the radial scaling of the background gauge field and conflicts with much of the allowed parameter space in (2.9) we focus on the negative root, which recovers $\zeta \to -\eta$ as $W_0 \to 0$. In terms of $W_0$ the parameter space (2.9) is

$$\left(0 < \eta \le \frac{1}{2}\left(\sqrt{5}-1\right) \text{ and } 0 < W_0 < \frac{1}{4}\left((1+\eta)^2 - (1-\eta-\eta^2)^2\right)\right)$$

or $\tag{2.11}$

$$\left(\eta > \frac{1}{2}\left(\sqrt{5}-1\right) \text{ and } 0 < W_0 < \frac{(1+\eta)^2}{4}\right).$$

We can now see in Figure 1 how $k_\star$ changes as a function of $W_0$. As expected, we see from the left plot that the Fermi surface is suppressed as $W_0$ increases.

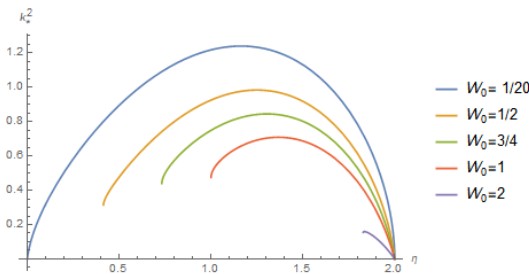

**Figure 1**: The critical momentum $k_\star^2$, below which low-energy spectral weight exists, as a function of $\eta$. The abrupt ending of the lines corresponds to the limit of our allowed parameter space.

## 2.2 Longitudinal Channel

In this channel, the low energy spectral weight

$$\sigma(k) = \lim_{\omega \to 0} \omega^{2\nu_- - 1} \tag{2.12}$$

becomes imaginary within a subregion of the allowed parameter space. This signals an instability, potentially toward a spatially modulated phase. We refer the reader to [12] for the exact form of $\nu_-$. The region of instability is

$$\left[0 < \eta \le \frac{1}{2}\left(\sqrt{5} - 1\right) \text{ and } 0 < \zeta < \frac{\eta^2}{2}\right] \text{ or } \left[\frac{1}{2}\left(\sqrt{5} - 1\right) < \eta < 1 \text{ and } 0 < \zeta < \frac{1-\eta}{2}\right]. \tag{2.13}$$

This region is plotted in terms of the broader allowed parameter space in Figure 2. Equation (2.13) basically restricts $\zeta < 0$. The instability region in terms of $W_0$ is

$$\left(0 < \eta \le \frac{1}{2}\left(\sqrt{5} - 1\right) \text{ and } \eta < W_0 < \frac{1}{4}\left((1+\eta)^2 - (1 - \eta - \eta^2)^2\right)\right)$$

or

$$\left(\eta > \frac{1}{2}\left(\sqrt{5} - 1\right) \text{ and } \eta < W_0 < \frac{(1+\eta)^2}{4}\right). \tag{2.14}$$

Equation (2.14) restricts $0 < W_0 < \eta$.

Figure 2 also depicts the region in which a Fermi shell exists, meaning a region of low-energy spectral weight over the range of momenta $k_- < k < k_+$, as was reported in [12]. This is the brown region in Figure 2. We are now ready to study how the size of the Fermi shell $\Delta k \equiv k_+ - k_-$ changes as a function of $W_0$. We obtain different qualitative results from those found in the transverse channel. That is, the size of the Fermi shell is increasing with increasing charge $W_0$, rather than decreasing. This is shown in Figure 3. We offer an interpretation for this in the Discussion. We note that $\Delta k$ is always finite within the brown stability region of Figure 2.

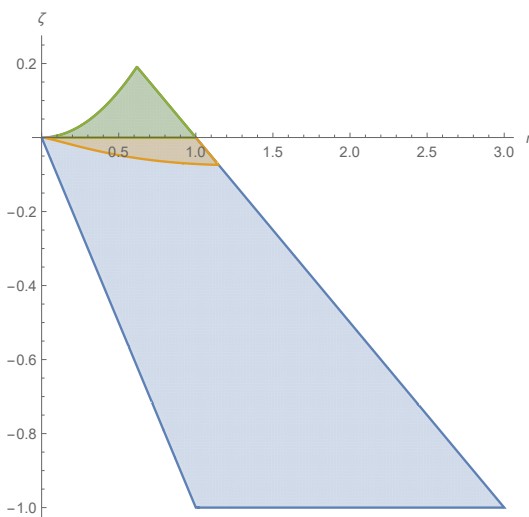

**Figure 2**: In the green region, the exponent $\nu_-(k)$ is complex for a range of $k$, signaling a finite $k$ instability. The exponent $\nu_-(k)$ is real in the brown and blue regions for all values of $k$. In the brown region, $2\nu^- - 1 < 0$ for a range of wavevectors $k_-^\star < k < k_+^\star$, signaling the presence of a Fermi shell. In the blue region $2\nu^- - 1 > 0$, and thus no spectral weight exists in this region.

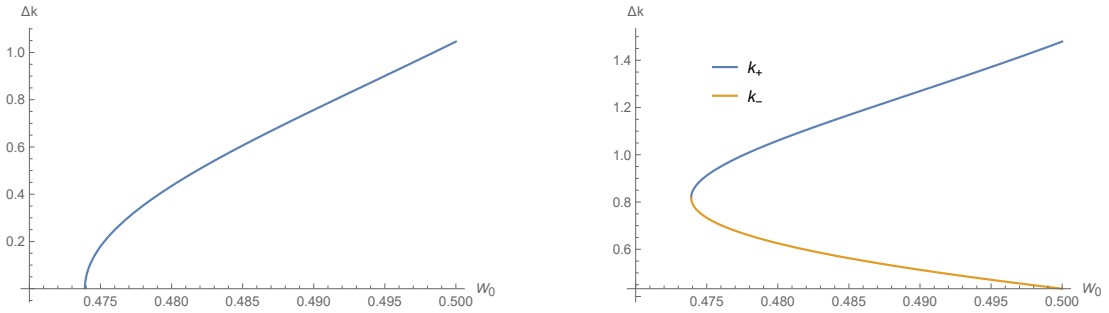

**Figure 3**: Left: The Fermi shell size $\Delta k \equiv k_+ - k_-$ is plotted as a function of condensate charge $W_0$, with $\eta = \frac{1}{2}$. Right: The critical momenta $k_+$ and $k_-$ are plotted separately, also with $\eta = \frac{1}{2}$. These figures capture our entire region of stability, namely $0 < W_0 < \eta$.

.

## 3 Einstein-Maxwell-dilaton with Axions

In this section, we study the impact of broken translational invariance alone on the low energy spectral weight by adding so-called axion terms to the Einstein-Maxwell-dilaton

theory. Specifically, we are interested in the following Lagrangian:

$$S = \int d^4x \sqrt{-g} \left[ R - \frac{1}{2}\partial\phi^2 - \frac{1}{4}Z(\phi)F^2 - \frac{1}{2}Y(\phi)\sum_i \partial\psi_i^2 - V(\phi) \right], \qquad (3.1)$$

where $i$ runs over boundary spatial dimensions (in our case two of them, $x$ and $y$). This theory was studied in [18] in the context of charge transport. To break translational invariance, we choose fields proportional to the coordinates

$$\psi_i = mx_i, \qquad (3.2)$$

and for simplicity we take the proportionality constant $m$ to be the same for each $x_i$. As before, we choose the following IR behavior that yields a scaling solution:

$$V(\phi) = V_0 e^{-\delta\phi}, \qquad Z(\phi) = Z_0 e^{\gamma\phi}, \qquad Y(\phi) = Y_0 e^{\lambda\phi}. \qquad (3.3)$$

For the rest of the analysis we are free to set $Z_0 = 1$ and $Y_0 = 1$. Our background parameters obey the following constraints:

$$A = \frac{\sqrt{2\eta - m^2 + 2}}{\eta + 1} r^{-\eta - 1}, \quad V_0 = -(\eta + 1)^2 - \frac{m^2}{2}, \quad \kappa = \sqrt{\eta(2 + \eta)}$$
$$\lambda = 0, \quad \kappa\delta = -\eta, \quad \kappa\gamma = \eta. \qquad (3.4)$$

The resulting parameter space for this theory is

$$\eta > 0 \qquad \text{and} \qquad -\sqrt{2 + 2\eta} < m < \sqrt{2 + 2\eta}. \qquad (3.5)$$

Radial deformations do not impose any further constraints on the parameter space.

### 3.1 Transverse Channel

We first consider the transverse channel, and include the following perturbations:

$$\{\delta A_y, \delta g_{ty}, \delta g_{ry}, \delta g_{xy}, \delta\psi_y\}. \qquad (3.6)$$

The $y$ in the scalar $\delta\psi_y$ is a distinguishing subscript and not meant to indicate a vector component. All perturbations take the plane wave form $\delta X = \delta X(r)e^{i(kx - \omega t)}$. We work in radial gauge $\delta g_{\mu r} = 0$. We wish to calculate the scaling exponent $\nu_-$ of the spectral weight:

$$\sigma(k) = \lim_{\omega \to 0} \frac{\text{Im}G^R_{JJ}(\omega, k)}{\omega} \propto \lim_{\omega \to 0} \omega^{2\nu_- - 1}. \qquad (3.7)$$

To achieve this, we define the following scaling behavior for the perturbations:

$$\delta A_y = a_0 r^{a_1}, \qquad \delta g_{ty} = t_0 r^{t_1}, \qquad \delta g_{xy} = x_0 r^{x_1}, \qquad \delta\psi_y = \psi_0 r^{\psi_1}. \qquad (3.8)$$

A scaling analysis of the perturbed equations of motion relate the above exponents, and the constants $x_0, \psi_0$ and $\psi_1$ drop out or decouple from the rest of the equations. Therefore, taking a coefficient array of the two remaining equations in terms of $a_0$ and $t_0$ and setting the determinant to zero allows us to solve for the radial scaling:

$$a_1 = \frac{1}{2}\left(1 - \eta \pm \sqrt{5 + \eta^2 + 2\eta + 4k^2 \pm 4\sqrt{\eta^2 + 2\eta(k^2 + 1) + k^2(2 - m^2) + 1}}\right).$$
(3.9)

We are interested in $\nu_-$, which is given by

$$2\nu_- = \sqrt{5 + \eta^2 + 2\eta + 4k^2 - 4\sqrt{\eta^2 + 2\eta(k^2 + 1) + k^2(2 - m^2) + 1}}.$$
(3.10)

This exponent is always real within our parameter space. This means that there is no instability in the transverse channel, which was also the case for the holographic superconductor. The critical wave number is found by solving the equation $2\nu_- - 1 = 0$ for $k$:

$$k_\star^2 = \frac{1}{8}\left(-2\eta^2 + 4\eta - 4m^2 \pm 2\sqrt{2}\sqrt{-4\eta^3 + 4\eta^2 + 6\eta + 2m^4 + (2\eta^2 - 4\eta + 1)m^2 - 2} - 1\right).$$
(3.11)

We can see that $k_\star$ vanishes at the values

$$\eta = \{-4, -2, 0, 2\}.$$
(3.12)

The parameter $\eta$ is constrained to be positive by the null energy condition, however.

In the transverse channel, we see that the larger $m$ gets, the more the spectral weight is suppressed. This is similar to the effect of the parameter $W_0$ that we saw previously for the holographic superconductor. Indeed, we will see just how similarly the effects of these two terms are on spectral weight in the next section. The spectral weight is never suppressed completely, as our parameter space (3.5) constrains us to consider only $|m| < \sqrt{2 + 2\eta}$.

## 3.2 Longitudinal channel

In the longitudinal channel, the perturbation variables are:

$$\{\delta A_t, \delta A_x, \delta g_{tt}, \delta g_{tx}, \delta g_{xx}, \delta g_{yy}, \delta \psi_x, \delta \phi\}.$$
(3.13)

We chose radial gauge $\delta g_{\mu r} = A_r = 0$. The modes $\delta A_x$ and $\delta g_{tx}$ decouple from the rest, and thus we can set them to zero.

As in the transverse channel, all perturbations take the plane wave form $\delta X = \delta X(r)e^{i(kx - \omega t)}$, and we define the scaling behavior for the perturbations as:

$$\delta A_t = a_0 r^{a_1}, \qquad \delta g_{tt} = t_0 r^{t_1}, \qquad \delta g_{xx} = x_0 r^{x_1},$$
$$\delta g_{yy} = y_0 r^{y_1}, \qquad \delta \psi_x = \psi_0 r^{\psi_1}, \qquad \delta \phi = \phi_0 r^{\phi_1}.$$
(3.14)

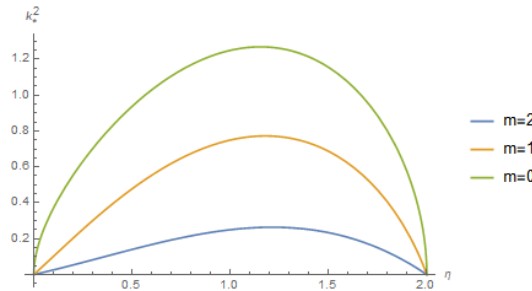

**Figure 4**: The critical momentum $k_\star^2$, below which low-energy spectral weight exists, as a function of $\eta$. This is for the massless vector ($W_0 = 0$) case, with axions ($m \neq 0$). Note that for the $m = 2$ curve, our allowed parameter space restricts us to $0 < \eta < 1$.

As before, we can use a scaling analysis to obtain the radial scaling of interest:

$$\nu_0 = \frac{1}{2}\left(\sqrt{(1+\eta)^2 + 4k^2 + 4m^2}\right)$$

$$\nu_\pm = \frac{1}{2}\sqrt{\frac{\left(\eta^3 + 12\eta^2 + 21\eta + 10 + 4k^2(\eta+2) - 2\eta m^2 \pm 2\sqrt{X}\right)}{(\eta+2)}}. \qquad (3.15)$$

where

$$X = 8k^2(\eta+1)(\eta+2)\left(2\eta - m^2 + 2\right) + \left(\eta\left(4\eta - m^2 + 8\right) + 4\right)^2. \qquad (3.16)$$

There are two major differences in the effects of broken $U(1)$ symmetry (as in the holographic superconductor) and broken translation invariance (as in the EMD plus axion theory) on the longitudinal channel. First, unlike for the holographic superconductor, here we find no instability in the longitudinal channel (i.e. $\nu_-$ is always real). Second, in the EMD plus axion case, there is no low energy spectral weight for any $m$. This generalizes the result found in [10] for the pure EMD theory in four dimensions.

## 4 Axion with Massive Vector

We are now ready to consider the Einstein-Maxwell-dilaton theory with a massive vector that breaks $U(1)$ symmetry and a massless scalar that breaks translation invariance:

$$S = \int d^4x \sqrt{-g}\left[R - \frac{1}{2}\partial\phi^2 - \frac{1}{4}Z(\phi)F^2 - \frac{1}{2}Y(\phi)\sum_i \partial\psi_i^2 - \frac{1}{2}W(\phi)A^2 - V(\phi)\right]. \qquad (4.1)$$

As in Section 3, we choose the axion ansatz $\psi_i = mx_i$, and the following IR scaling behavior for the action:

$$V(\phi) = V_0 e^{-\delta\phi}, \qquad Z(\phi) = Z_0 e^{\gamma\phi}, \qquad W(\phi) = W_0 e^{\epsilon\phi}, \qquad Y(\phi) = Y_0 e^{\lambda\phi}. \qquad (4.2)$$

Henceforth we set $Z_0 = 1$ and $Y_0 = 1$.

Our metric and fields take the form:

$$ds^2 = r^{-\eta}\left(\frac{-dt^2 + dr^2}{r^2} + dx^2 + dy^2\right), \qquad A = A(r)\mathrm{d}t, \qquad \phi(r) = \kappa\log r \qquad (4.3)$$

and our background parameters obey the constraints:

$$A = \sqrt{\frac{m^2 - 2(\eta + 1)}{(\zeta - 1)(\eta + 1)}}r^{\zeta - 1}, \qquad \kappa = \sqrt{\frac{\zeta\left(m^2 - 2(\eta + 1)\right) + \eta\left(\eta^2 + \eta + m^2\right)}{\eta + 1}}$$

$$V_0 = -\frac{2(\eta + 1)\left(-\zeta + \eta^2 + \eta + 1\right) + m^2(\zeta + 2\eta + 1)}{2(\eta + 1)}, \qquad \kappa\delta = -\eta, \qquad \kappa\gamma = \eta, \qquad (4.4)$$

$$W_0 = (1 - \zeta)(\zeta + \eta), \qquad \lambda = 0.$$

The parameter space arising from the reality of these background quantities, imposing $V_0 < 0$ and $W_0 > 0$, and from the null energy condition is

| $0 < \eta < \sqrt{2}$ | $-\eta < \zeta \leq \frac{\eta^2}{2}$ | $-\sqrt{2 + 2\eta} \leq m \leq \sqrt{2 + 2\eta}$ | |
| --- | --- | --- | --- |
| | $\frac{\eta^2}{2} < \zeta < 1$ | $-\sqrt{2 + 2\eta} \leq m < -\sqrt{\frac{(1+\eta)(2\zeta - \eta^2)}{\zeta + \eta}},$ | $\sqrt{\frac{(1+\eta)(2\zeta - \eta^2)}{\zeta + \eta}} < m \leq \sqrt{2 + 2\eta}$ |
| $\sqrt{2} \leq \eta$ | $-\eta < \zeta < 1$ | $-\sqrt{2 + 2\eta} \leq m \leq \sqrt{2 + 2\eta}$ | |

This is not the full parameter space, however. We must also consider radial deformations to the background (4.3) of the form

$$ds^2 = -D(r)dt^2 + B(r)dr^2 + C(r)(dx^2 + dy^2), \qquad A = \tilde{A}(r)\mathrm{d}t, \qquad \phi(r) = \tilde{\phi}(r) \qquad (4.5)$$

with

$$D(r) = r^{-\eta - 2}(1 + \epsilon D1 r^\beta), \quad B(r) = r^{-\eta - 2}(1 + \epsilon B1 r^\beta), \quad C(r) = r^{-\eta}(1 + \epsilon C1 r^\beta),$$
$$\tilde{A}(r) = r^{\zeta - 1}(1 + \epsilon A1 r^\beta), \qquad \tilde{\phi}(r) = \log(r^\kappa(1 + \epsilon\phi 1 r^\beta)) \qquad (4.6)$$

and $D1$, $B1$, $C1$, $A1$, $\phi1$ are constants. There are three pairs of radial deformations, each pair summing to $1 + \eta$. One of the pairs is just $(0, 1 + \eta)$, while the other two are

$$\beta_{\pm,\pm} = \frac{1}{2}\left(1 + \eta \pm \sqrt{\frac{A \pm 2C}{(\eta + 1)^2 S}}\right), \tag{4.7}$$

where $A(\eta, \zeta, m)$, $B(\eta, \zeta, m)$ and $S(\eta, \zeta, m)$ are given in the Appendix A. Note that in the case of the holographic superconductor, the mode $(0, 1 + \eta)$ is doubly degenerate. That is, we had the freedom to write two of the constants (say $D1$ and $\phi1$) in terms of the other three. In particular, $C1$ was a free parameter. The axion term forbids us from choosing $C1$ independently of the other constants. This is because our ansatz $\psi = mx$ should be kept fixed. One might imagine that one could simply undo a rescaling of $x$ with an appropriate rescaling of $m$, but because the other constants $D1$, etc depend on $m$, this is not an independent rescaling. To analyze the parameter space resulting from these deformations, we first need to ensure that all of the $\beta$s are real. We then require that we have two irrelevant modes (corresponding to $\beta < 0$). There are only two modes that have a possibility of being negative, namely $\beta_{-+}$ and $\beta_{--}$. Since $\beta_{-+} < \beta_{--}$, it is enough to require that $\beta_{--} < 0$. The resulting parameter space is too complicated to write down in closed form, but a portion of it is rendered in Figure 5. This can be compared with the parameter space for the holographic superfluid ($m = 0$) given in Figure 2. We see that the effect of $|m| > 0$ is to increase our allowed parameter space to include larger positive values of $\zeta$ (although the bound $\zeta < 1$ reported in the table above still holds).

## 4.1 Transverse Channel

We first consider the transverse channel, with the following perturbations:

$$\{\delta A_y, \delta g_{ty}, \delta g_{xy}, \delta \psi_y\}. \tag{4.8}$$

Again, the $y$ subscript in the scalar perturbation $\delta\psi_y$ is just a distinguishing subscript and not a vector index. All perturbations take the plane wave form $\delta X = \delta X(r)e^{i(kx - \omega t)}$. We endow the perturbations with scaling profiles:

$$\delta A_y = a_0 r^{a_1}, \qquad \delta g_{ty} = t_0 r^{t_1}, \qquad \delta g_{xy} = x_0 r^{x_1}, \qquad \delta \psi_y = \psi_0 r^{\psi_1}. \tag{4.9}$$

Redoing the scaling analysis of Section 3.1, we obtain the scaling exponent for the holographic superfluid with broken translational symmetry:

$$a_1 = \frac{1}{2}\left(1 + 2\zeta + \eta \pm \sqrt{\eta^2 + 2\eta + 4k^2 + 5 + \frac{2m^2(\zeta + \eta) \pm 2X_1}{\eta + 1}}\right) \tag{4.10}$$

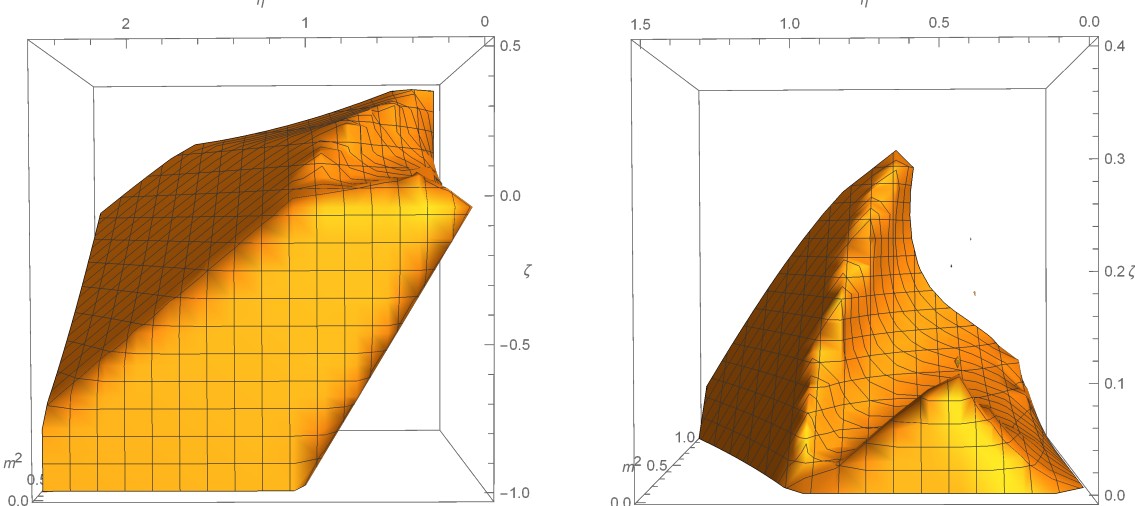

**Figure 5**: **Left:** A representative portion of the allowed parameter space for the EMD theory with broken $U(1)$ and translation symmetries. **Right:** A subregion of the allowed parameter space (with $0 < m^2 < 1$) that will be useful for comparisons below.

where

$$X_1 = \sqrt{(m^2(\zeta + \eta) - 2(1+\eta)^2)^2 - 4k^2(1+\eta)(\zeta - 1)(2 + 2\eta - m^2)} \qquad (4.11)$$

which gives

$$2\nu_- = \sqrt{\eta^2 + 2\eta + 4k^2 + 5 + \frac{2m^2(\zeta + \eta) - 2X_1}{\eta + 1}}. \qquad (4.12)$$

The exponent $\nu_-$ is always real within our parameter space, signaling again that there are no instabilities in this channel. When $m = 0$ we reproduce the result obtained in [12]. The low energy spectral weight for the transverse channel is thus

$$\sigma(k) = \lim_{\omega \to 0} \frac{\mathrm{Im}G_{JJ}^R(\omega, k)}{\omega} = \begin{cases} \infty & k < k_\star \\ 0 & k > k_\star \end{cases} \qquad (4.13)$$

where

$$k_\star^2 = \frac{1}{4}\left(-4\zeta - \eta(\eta + 2) - 2m^2 + 2\sqrt{2\left(2\zeta^2 + \eta(\zeta + 1)(\eta + 2)\right) + Y_m}\right) \qquad (4.14)$$

and

$$Y_m = m^4 - \frac{m^2(\eta(\zeta + 3)(\eta + 2) + 4\zeta)}{\eta + 1}. \qquad (4.15)$$

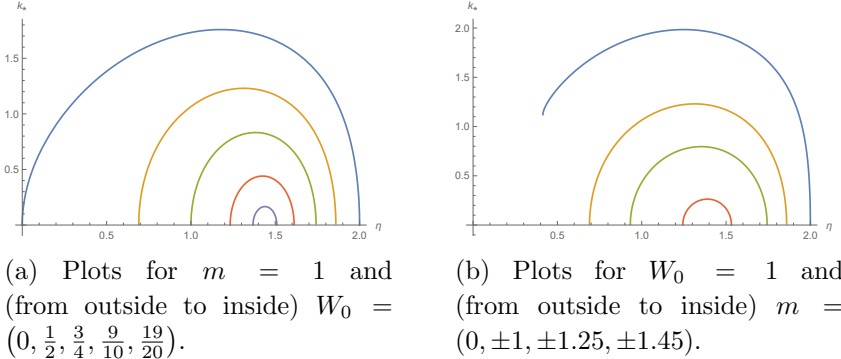

(a) Plots for $m = 1$ and (from outside to inside) $W_0 = \left(0, \frac{1}{2}, \frac{3}{4}, \frac{9}{10}, \frac{19}{20}\right)$.

(b) Plots for $W_0 = 1$ and (from outside to inside) $m = (0, \pm 1, \pm 1.25, \pm 1.45)$.

**Figure 6**: The Fermi surface size in the transverse channel $k_\star$ as a function of $\eta$. Only the $\zeta_-$ root yields real results.

Unlike the holographic superfluid case [12], a nonzero axion term forbids $k_\star$ from vanishing at $\eta = 2$. However, it does vanish at the special value

$$\zeta = \frac{\eta(2 - 4m^2 + \eta - \eta^2)}{4m^2}, \tag{4.16}$$

which is nonzero inside the parameter space.

In Figure 6 we again see that the axion term and the vector mass term affect the critical momentum $k_\star$ in much the same way, that is to suppress low-energy spectral weight as their magnitudes grow. Indeed, the effect of one term barely seems to influence the other: the two effects do not appreciably mix in this channel. The reader will find that this is not the case in the longitudinal channel, however.

## 4.2 Longitudinal channel

Now we turn to the longitudinal channel. The perturbation variables are:

$$\{\delta A_t, \delta A_x, \delta g_{tt}, \delta g_{tx}, \delta g_{xx}, \delta g_{yy}, \delta \psi_x, \delta \phi\} \tag{4.17}$$

The modes $\delta A_x$ and $\delta g_{tx}$ decouple from the rest, and thus we can set them to zero.

As in the transverse channel, all perturbations take the plane wave form $\delta X = \delta X(r)e^{i(kx - \omega t)}$, and we define the scaling behavior for the perturbations as:

$$\delta A_t = a_0 r^{a_1}, \qquad \delta g_{tt} = t_0 r^{t_1}, \qquad \delta g_{xx} = x_0 r^{x_1},$$
$$\delta g_{yy} = y_0 r^{y_1}, \qquad \delta \psi_x = \psi_0 r^{\psi_1}, \qquad \delta \phi = \phi_0 r^{\phi_1}. \tag{4.18}$$

As before, we can use a scaling analysis to obtain the radial scaling of interest. Setting $m = 0$ reproduces the result found in [12].

For the longitudinal channel we again expect three scaling exponents: $\nu_0$ and $\nu_\pm$. In this case the closed form of the $\nu$ exponents are too complicated to report here, but they are of the form:

$$\nu_{Y_i} = \frac{1}{2}\left(\sqrt{(1+\eta)^2 + 4k^2 + Y_i}\right) \tag{4.19}$$

where the $Y_i$ are solutions to the cubic equation

$$aY_i^3 + bY_i^2 + cY_i + d = 0, \tag{4.20}$$

with

$$
\begin{aligned}
a =&(\eta+1)^2\left(\zeta\left(m^2 - 2(\eta+1)\right) + \eta\left(\eta^2 + \eta + m^2\right)\right)\\
b =& -16\zeta^3(\eta+1)^2\left(m^2 - 2(\eta+1)\right) - 8\zeta^2(1+\eta)\left(m^2 - 2(\eta+1)\right)\left((\eta+1)(2\eta-3) + m^2\right)\\
& -4\zeta(1+\eta)\left(3\eta^3 + \eta\left(4m^2 - 1\right) + 2\right)\left(m^2 - 2(\eta+1)\right) - 8\eta(1+\eta)\left(\eta\left(m^2 - \eta\right) + 1\right)\left(\eta^2 + \eta + m^2\right)\\
c =& -32\left(m^2 - 2(\eta+1)\right)(\zeta-1)(\eta+1)^2 k^2\left(2\zeta\eta + \zeta(2\zeta-1) + \eta^2\right)\\
& +16\left(m^2 - 2(\eta+1)\right)(\eta+1)m^2(\zeta+\eta)\left((\eta+1)\left(2\zeta^2 + 2\zeta(\eta-1) + (\eta-1)\eta\right) - (\zeta-1)k^2\right)\\
& +16\left(m^2 - 2(\eta+1)\right)m^4(\zeta+\eta)^3\\
d =& -64k^2m^2(1-\zeta)(\zeta+\eta)^2\left(m^2 - 2\eta - 2\right)\left(2\eta^2 + 4\eta + m^2 + 2\right).
\end{aligned}
\tag{4.21}
$$

It is the $d$ term in (4.20) that complicates the scaling exponent solution substantially compared with our previous cases. We see that $d$ depends upon both of our main parameters of interest: the translation-breaking axion parameter $m$ and the condensate charge $W_0 = (1-\zeta)(\zeta+\eta)$. Unlike in the transverse channel, here our scaling exponents $\nu_{Y_i}$ depend heavily on the how the effects of the axion and the condensate terms act together.

Nevertheless, we can still analyze the spectral weight in this channel numerically. We begin by determining how the instability region for the holographic superfluid reported in Figure 2 changes in the presence of a symmetry breaking axion term. This new instability region is presented in Figure 7. We see that the axion strength $m$ allows for a larger viable parameter space (as reported in Figure 5) and thus an augmented instability region is possible. Note however that this instability region still only exist for $\zeta > 0$, as was the case for the holographic superfluid. Unlike for the holographic superfluid, though, we now see that the stable region is not restricted to $\zeta < 0$. The new stability region that exists for $\zeta > 0$, which is partially depicted in Figure 7, grows steeply with increasing $|m|$.

We now turn to the question of whether there exists low-energy spectral weight at finite momentum $k$ in the longitudinal channel, either in the form of a smeared

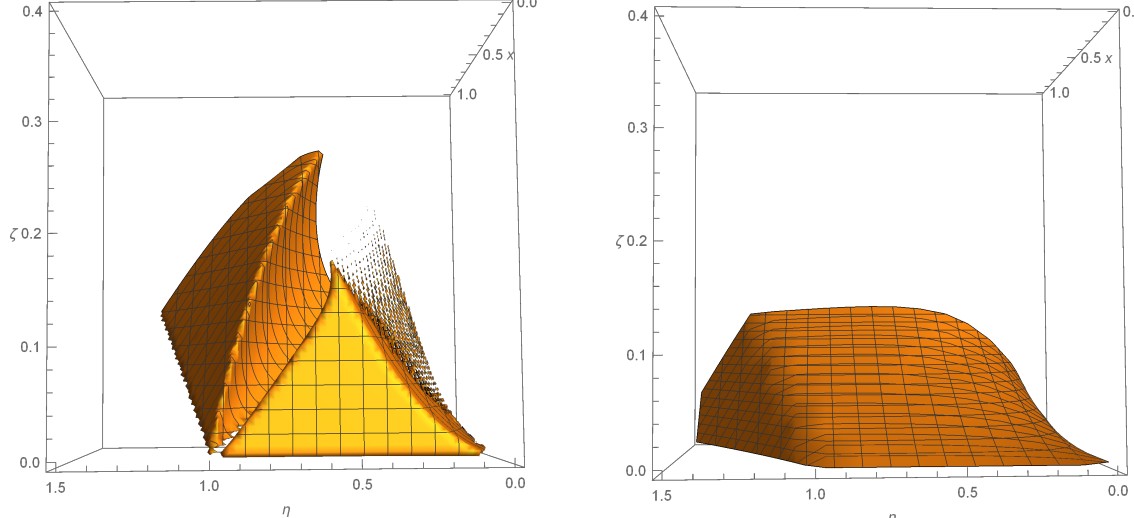

**Figure 7**: **Left:** The instability region for the EMD theory with $U(1)$ and translational symmetries broken. Here $x = m^2$. The presence of the axion strength $m$ allows for an augmented parameter space, and thus a richer instability structure. **Right:** A new stability region that exists for $\zeta > 0$, appearing for $m \neq 0$.

Fermi surface or a Fermi shell. As before, the condition for nonzero spectral weight is $2\nu_- - 1 < 0$. We will begin by presenting our results for the spectral weight in the presence of both the translation symmetry breaking axion term and the $U(1)$ symmetry breaking massive vector, and then compare these results to those presented for the holographic superfluid (in Section 2.2 and in [12]) and for the axion alone (in Section 3.2). The main results for the holographic superfluid in Section 2.2 that we would like to keep in mind are:

1. A finite $k$ instability appears for $\zeta > 0$, effectively restricting our analysis of low-energy spectral weight to the region $\zeta < 0$ (2.13).

2. For an appropriate region of the parameter space (Figure 2) we see a Fermi shell, rather than a smeared Fermi surface.

3. The Fermi shell width ($\Delta k \equiv k_+ - k_-$) increases with decreasing charge $W_0$.

The main results for the EMD plus axion theory in Section 3.2 to remember are:

(i) In the longitudinal channel there is no spectral weight for any $m$.

(ii) All values of $m$ in the region $0 < |m| < \sqrt{2 + 2\eta}$ are allowed.

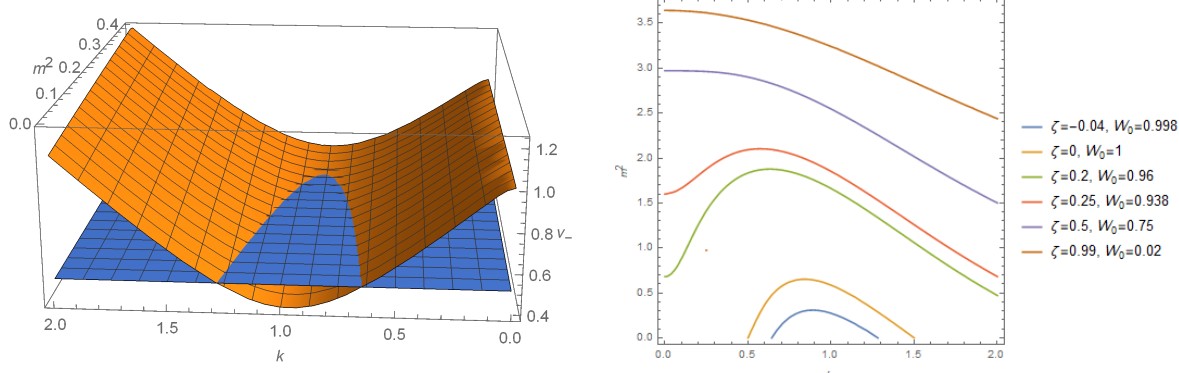

**Figure 8**: Low-energy spectral weight results for $\eta = 1$. **Left:** Non-zero low-energy spectral weight corresponds to the exponent $\nu_-$ dipping below the $\nu_- = 1/2$ plane. In this plot $\zeta = -.04$. For small $|m|$ we have a Fermi shell, and for large enough $|m|$ spectral wieght is suppressed. **Right:** Contour plots of the intersection of $\nu_-$ with the $\nu_- = 1/2$ plane for various values of $\zeta$.

The results for the longitudinal low-energy spectral weight (for the representative value $\eta = 1$) are presented in Figure 8. Non-zero spectral weight corresponds to the scaling exponent $\nu_-$ dipping below the $\nu_- = 1/2$ plane. For negative $\zeta$ there are two distinct regions of interest. For small enough $\zeta$ (approximately between $-1 < \zeta < -.07$ for $\eta = 1$) there is no spectral weight for any $m$. This generalizes the result (i) above (which corresponded to $\zeta = -1$, since $\eta = 1$ and $\zeta = -\eta$ means $W_0 = 0$) to a range of $\zeta$. For larger $\zeta$, in the approximate region $-.07 < \zeta < 0$, we have a Fermi shell (as in point 2) that increases in size with decreasing $W_0$ (as in point 3) but decreases with increasing $m$. Comparing with Figure 2 for the holographic superfluid, we see that the presence of the axion parameter $m$ does not significantly affect the parameter space region that supports low-energy spectral weight, despite the fact that increasing $m$ decreases the shell width.

In the holographic superfluid, positive $\zeta$ was not allowed due to the finite $k$ instability (point 1). However, the axion term allows for stable theories with positive $\zeta$, the price being that not all $m$ in the region given in point (ii) are allowed. For some values of $\zeta > 0$ the spectral weight is still a Fermi shell, but when $\zeta$ gets large enough our contour becomes monotonic in $k$, and we have a smeared Fermi surface.

## 5 Discussion

Here we have examined the low-energy spectral weight and stability structure of three bottom-up models: the holographic superfluid characterized by broken $U(1)$ symmetry,

the EMD plus axion theory which spontaneously breaks translation symmetry, and the holographic superfluid plus axion theory in which both symmetries are broken. We find that the results for the transverse channels of these theories are largely the same. There is never any instability in the transverse channel, and there is always a smeared Fermi surface. We also find that the condensate charge $W_0$ and the axion strength $m$ have the same effect: the Fermi surface size $k_\star$ decreases with increasing $W_0$ and $m$. As discussed in Section 2, this aligns with the naïve intuition that it should be easier for the scalar to condense at large charge.

The longitudinal channels give more diverse results. In the EMD plus axion theory of Section 3.2, there is no low-energy spectral weight for any $m$ (though this restriction may be lifted when considering a higher number of spacetime dimensions; see [7] for an example). There is also no instability in this theory for any $m$. Thus it is the $U(1)$ symmetry breaking term $W_0$ that drives both the existence of Fermi shells and the presence of an instability at finite momentum $k$. However, once these phenomena are present, the axion strength $m$ affects the structure of the spectral weight and the instability region, as seen by comparing the results of Sections 2.2 and 4.2. Namely, increasing $|m|$ augments the instability region that was present for the holographic superfluid to include $\zeta > 0$ (Figure 7) and suppresses low-energy spectral weight for each $\zeta$ (Figure 8).

Note that our expectation that large charge $W_0$ should facilitate condensation, and thus shrink the size of the Fermi surface, was not borne out in the longitudinal channels of Sections 2.2 and 4.2 where Fermi shells are present. That is, we note from Figure 3 that only $k_+$ increases with $W_0$, while $k_-$ decreases as was naïvely anticipated. One possible explanation for this lies in fact that we think of these Fermi shells (or nested Fermi surfaces) as *smeared*. This is is contrast to the sharply defined Fermi surface that exists for free fermions at zero temperature. Perhaps this smearing is telling us that it is some intermediate value of $k$ between $k_+$ and $k_-$ that is of true physical interest. Consider Figure 8, for example. While it's true that the Fermi shell width $\Delta k = k_+ - k_-$ increases with each increasing $\zeta$ curve, the peak of each $\zeta$ curve shifts to the left, as one might anticipate according to the discussion above. In future work it will be desirable to formulate a connection between bottom-up models exhibiting Fermi shells and the top-down constructions containing Fermi shells, such as those mentioned in the Introduction [15, 16].

## Acknowledgments

We would like to thank Blaise Goutéraux in particular for many useful discussions and contributions regarding this work. We also thank Sean Hartnoll for insightful comments

and Nikhil Monga for helpful contributions.

# A    Supplemental material

$$A = B + 2\zeta\eta R\left(3\eta^2 + 2\eta + 2m^2 - 1\right) + S\left(\eta\left((\eta - 1)\eta + 4m^2 + 3\right) + 5\right) \tag{A.1}$$

$$B = 4\zeta^2 R\left((\eta + 1)(2\eta - 3) + m^2\right) + 8\zeta^3(\eta + 1)R \tag{A.2}$$

$$C = \sqrt{D^2 - 4m^2 RS(\zeta + \eta)\left(\zeta^2\left(2(\eta + 1)^2 + m^2\right) + (2\zeta + \eta)(-\eta - \zeta R + S - 1)\right)} \tag{A.3}$$

$$D = \left(\frac{B}{2} + 2\eta\left(\eta\left(m^2 - \eta\right) + 1\right)\left(\eta^2 + \eta + m^2\right) + \zeta R\left(3\eta^3 + \eta\left(4m^2 - 1\right) + 2\right)\right) \tag{A.4}$$

$$R = \left(m^2 - 2(\eta + 1)\right) \tag{A.5}$$

$$S = \left(\zeta R + \eta\left(\eta^2 + \eta + m^2\right)\right). \tag{A.6}$$

# B    Review of spectral weight

## B.1    What is the spectral weight?

Here we motivate the quantity that we are calculating, the spectral weight:

$$S(k) = \frac{\mathrm{Im}G^R_{\mathcal{OO}}(\omega, k)}{\omega}. \tag{B.1}$$

We reserve the symbol $\sigma$ to denote the *low energy* spectral weight:

$$\sigma(k) = \lim_{\omega \to 0} \frac{\mathrm{Im}G^R_{\mathcal{OO}}(\omega, k)}{\omega}. \tag{B.2}$$

Possible operators of interest are $\mathcal{O} = J^t$, in which case $G^R_{J^t J^t}(\omega, k)$ is the density-density correlation function, and $\mathcal{O} = J^x$, in which case $G^R_{J^x J^x}(\omega, k)$ is a current-current correlator. In a fermionic theory with $\mathcal{O} = \psi$, the Green's function is the fermion propagator, and a Fermi surface corresponds to a pole in this quantity at the Fermi momentum $k = k_F$. In this work we compute the Green's function for generalized current operators $\mathcal{O} = J^{\parallel}$ and $\mathcal{O} = J^{\perp}$.

## B.2    What does ARPES measure?

In this subsection we follow the discussion presented in [6, 28]. Angle-resolved photoemission spectroscopy (ARPES) is a measurement technique that directly probes the distribution of electrons in a medium. That is, by ejecting electrons from a sample,

ARPES measures the density of single-particle electron excitations governed by the fermion propagator $G^R_{\psi\psi}(\omega, k)$, or more directly the *single-particle spectral function*

$$A(\omega, k) \equiv -\frac{1}{\pi} \text{Im} G^R_{\psi\psi}(\omega, k). \tag{B.3}$$

A pole in the spectral function $A(\omega, k)$ as $\omega \to 0$ signifies the presence of a Fermi surface. This is immediately clear in the case of free fermions, where the propagator is

$$G^R_{\psi\psi} = \frac{1}{\omega - \xi(k) + i\epsilon}, \tag{B.4}$$

where

$$\xi(k) = \frac{k^2}{2m} - \mu = \frac{k^2}{2m} - \frac{k^2_F}{2m} = v_F(k - k_F). \tag{B.5}$$

By examining equation (B.4), we see that the low energy pole occurs at $k = k_F$. The correspondence between a pole in $G^R_{\psi\psi}$ and the existence of a Fermi surface also exists in interacting theories (even strongly interacting theories), in which the propagator becomes

$$G^R_{\psi\psi} = \frac{Z}{\omega - v_F(k - k_F) + \Sigma(\omega, k)}. \tag{B.6}$$

In (B.6), $Z$ is called the quasi-particle weight and

$$\Sigma(\omega, k) = \frac{i\Gamma}{2} \tag{B.7}$$

is the self-energy, with $\Gamma$ the particle decay rate. In fact, experiments have shown that (B.6) is the form that the propagator takes in the now famous "strange metal" phase of certain high $T_c$ cuprate superconductors [29], with

$$\Sigma(\omega) = C\omega \log \omega + D\omega, \tag{B.8}$$

where $C$ is real and $D$ is complex. This matches a theoretical model known as a *marginal Fermi liquid* [2]. For clarity, the scaling of the imaginary part of the self-energy with $\omega$ for various theories is given in Table 1.

### B.3   What do we measure in this paper?

In holographic calculations, there are at least two distinct ways to search for the presence of a Fermi surface (or, more generally, the presence of Pauli exclusion). The first method is to directly compute the single-particle spectral function $A(\omega, k)$ in the bulk and see if it has a pole at some momentum $k_F$ as $\omega \to 0$. Calculating $A(\omega, k)$ requires knowledge of "UV" or near-boundary data ($G^R_{\psi\psi}$ is the UV propagator), and so in practice one must

| Fermi liquid | $\mathrm{Im}\Sigma(\omega) \sim \omega^2$ |
|---|---|
| Semi-local quantum liquid | $\mathrm{Im}\Sigma(\omega) \sim \omega^{2\nu_k}$ |
| Strange metal (marginal Fermi liquid, $\nu_k = 1/2$) | $\mathrm{Im}\Sigma(\omega) \sim \omega.$ |

**Table 1**: The scaling of the imaginary part of the self-energy for Fermi liquid theory, the semi-local quantum liquid, and the marginal Fermi liquid. The exponent $\nu_k$ is related to the conformal dimension of the dual operator by $\delta_k = \nu_k + \frac{1}{2}$.

1. Consider a theory with at least one bulk fermion $\psi$.

2. Linearly perturb the bulk fields (for example $\psi \to \psi + \delta\psi$).

3. Solve the Dirac equation for the perturbed fields over the entire spacetime (this can be done numerically if necessary).

4. Read off the IR propagator via the standard holographic relationship

$$G_{\psi\psi}^R(\omega, k) \propto \frac{\psi_{(1)}}{\psi_{(0)}}, \tag{B.9}$$

where $\psi_{(0)}$ and $\psi_{(1)}$ are obtained from the near boundary expansion of the perturbed field

$$\delta\psi(z \to 0) = \frac{\psi_{(0)}}{L^{d/2}} z^{d-1-\Delta_k} + ... + \frac{\psi_{(1)}}{L^{d/2}} z^{\Delta_k} \tag{B.10}$$

for a $d+2$-dimensional bulk spacetime. $L$ is the AdS radius, and $\psi_{(0)}$ and $\psi_{(1)}$ are constants in the radial coordinate $z$ but depend upon $\omega$ and $k$ (see for example [4] for a review of these concepts).

This was the approach taken in [30–33].

The second method differs from the preceding one in several ways. First, we do not include any explicit bulk fermions $\psi$. Second, instead of looking at propagators of our bulk fields, we are interested in more general correlation functions $G_{\mathcal{O}\mathcal{O}}^R(\omega, k)$ and their associated low energy spectral weight

$$\sigma(k) = \lim_{\omega \to 0} \frac{\mathrm{Im} G_{\mathcal{O}\mathcal{O}}^R(\omega, k)}{\omega}. \tag{B.11}$$

The operators $\mathcal{O}$ that we consider are related for example to charge density $J^t$ and current $J^x$, but are not exactly these. Rather, we study operators that we can call $J^\parallel$ and $J^\perp$, arising from the decoupling of the perturbed fields into transverse and longitudinal channels. Finally, we restrict ourselves to the near-horizon IR geometry.

We will always call the associated IR Green's function $\mathcal{G}^R_{\mathcal{O}\mathcal{O}}$ to differentiate it from the UV one. In fact, at low energies (that is, $\omega << \mu$) the IR and UV Green's functions can be related through a matching argument [4]:

$$G^R_{\mathcal{O}\mathcal{O}}(\omega, k) = \frac{b^1_{(1)} + b^2_{(1)}\mathcal{G}^R_{\mathcal{O}\mathcal{O}}(\omega, k)}{b^1_{(0)} + b^2_{(0)}\mathcal{G}^R_{\mathcal{O}\mathcal{O}}(\omega, k)} \tag{B.12}$$

where the $b$'s are real constants independent of $\omega$. On the right hand side of (B.12), all of the UV data is stored in the real constants. Taking the imaginary part of (B.12), we find, to leading order as $\omega \to 0$ [6],

$$\mathrm{Im}G^R_{\mathcal{O}\mathcal{O}}(\omega, k) \propto \frac{\mathrm{Im}\mathcal{G}^R_{\mathcal{O}\mathcal{O}}(\omega, k)}{(b^1_{(0)})^2}. \tag{B.13}$$

We have kept the real constant explicit in (B.13) rather than folding it into the proportionality to make a point. If the constant $b^1_{(0)} = 0$, then we get a pole in the spectral function $A(\omega, k) \sim \mathrm{Im}G^R_{\mathcal{O}\mathcal{O}}$, and this would indicate the presence of a Fermi surface. For our purposes, we are only calculating $\mathrm{Im}\mathcal{G}^R_{\mathcal{O}\mathcal{O}}$, and so we do not have access to the UV data and thus cannot determine whether $A(\omega, k))$ possesses such a pole. *Nevertheless*, it turns out that there is a second indicator of a Fermi surface and Pauli exclusion apart from this pole. We now describe how this works.

The spectral weight $\sigma(k)$ is aptly named, as it admits a spectral decomposition [4]:

$$\mathrm{Im}G^R_{JJ}(\omega, k) = \sum_{m,n} e^{-\beta E_m} \left| \langle n(k') | J(k) | m(k'') \rangle \right|^2 \delta(\omega - E_m + E_n). \tag{B.14}$$

The sums in (B.14) are sums over eigenstates. There are actually two delta functions in (B.14), one in the energy difference between states and one in the momentum difference, resulting from the inner product. The $J$ tells us, then, that the spectral weight counts *charged* degrees of freedom that exist at a given frequency and momentum. Therefore, if one takes the $\omega \to 0$ limit of (B.14) and finds that there are low energy degrees of freedom at non-zero $k$, one can conclude that the charged particles have not condensed, and a phenomenon resembling Pauli exclusion is at work.

If we again take $\mathcal{O} = J$, then the spectral weight is also the real part of the electrical conductivity (see for example [34]). One can see this by comparing Ohm's law[4]

$$J(\omega) = \tilde{\sigma}(\omega)E(\omega) \tag{B.15}$$

---

[4]The tilde over the conductivity is simply to differentiate it from the spectral weight, which is also referred to as $\sigma$ in the literature.

to the linear response expression[5]

$$\langle J(\omega) \rangle = G_{JJ}^R(\omega) A(\omega) = \frac{G_{JJ}^R(\omega)}{i\omega} i\omega A(\omega) = \frac{G_{JJ}^R(\omega)}{i\omega} E(\omega). \tag{B.16}$$

From (B.15) and (B.16), we can see that

$$\tilde{\sigma} = \frac{G_{JJ}^R(\omega)}{i\omega} \tag{B.17}$$

This motivates the division by $\omega$ in the definition of the spectral weight, and from (B.17) we also see that $\mathrm{Re}\tilde{\sigma}(\omega) = \mathrm{Im}G_{JJ}^R(\omega)$.

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
