# Peer review of "Spectral weight in holography with momentum relaxation"

_SciPost Physics_

## Round 1 · Referee Report · Anonymous (Referee 1) · 2022-7-20

Report

The author investigates a family of bottom-up holographic models of a condensed matter system displaying both explicit translation symmetry breaking and spontaneous U(1) symmetry breaking. In particular, she investigates the imaginary part of the current-current two point function (for the U(1)) in momentum space in order to infer properties of the corresponding spectral density.
Section 2 is largely a review of [12], where the author and a collaborator considered the same system in the absence of translation symmetry breaking.
Thus the new results concern the effect of translation symmetry breaking. The story appears to be qualitatively similar.
Increasing the charge of the condensate appears to decrease the size of the Fermi sea in the transverse channel, while it increases the size of a Fermi
shell in the longitudinal channel, even in the presence of translation symmetry breaking. Considered on its own, the translation symmetry breaking parameter
suppresses the Fermi sea/shell both in the transverse and longitudinal channels.

There is also a second story which concerns the appearance of an instability at finite momenta in the U(1) broken system, suggesting presumably that there is a region in parameter space where the solution considered is not the true ground state, and the system left to its own should spontaneously break translation symmetry. Apparently in the presence of explicit translation breaking, this instability is enhanced, ruling out a larger swath of the parameter space.

The manuscript is clear and well written. The results are interesting, and the system is one of current interest in the AdS/CMT community.
I recommend publication.

A comment is that sometimes interactions can make bosons look like fermions. A famous example is the Lieb-Liniger model in the limit of infinite coupling, where a sort of Pauli exclusion take place not because the objects are fermions but because the bosonic particles are physically repelled from each other through the interaction.

Requested changes

A weakness of the manuscript is a lack of physical interpretation. I think more work could have been done to either try to make a connection to actual condensed matter systems, or to explain where the problems are in making the connection more precise.
For example, I have a basic puzzle about why the structure of the Fermi surface should look different in the longitudinal and transverse channels. I would have thought that if there really are fermions in the dual field theory, the same fermions should contribute both to the transverse and longitudinal spectral density. And yet the results here suggest otherwise. Why is the spectral density in the longitudinal channel at zero momentum suppressed compared to the transverse one? Is there a plausible physical mechanism that can explain that suppression?

I was also puzzled about the ``intuition'' that increasing the charge should make the Fermi surface smaller. Is there something like this that happens in BCS theory? Where does this intuition come from?

If the author could add a little more connective tissue tieing her results to condensed matter systems, I would be grateful.

---

## Round 1 · Referee Report · Anonymous (Referee 2) · 2022-9-5

Report

I agree generally with the remarks of the previous referee. A better physical explanation would be useful.

A few comments on the presentation: 1. $\mathcal{G}$ is not defined in (1.2), and one has to read through all of Appendix B to figure out what it is. 2. Above (1.4), some comments on how this represents a superconductor would be helpful i.e. \phi is the amplitude of the order parameter, and the phase is ignored. 3. Actually I am puzzled why the phase of the order parameter does not need to be considered in computing the conductivity. 4. The $\psi_i x_i$ above (1.4) appears to be a typo. 5. From (3.1) and (3.2) it is not clear why $Y(\phi)$ cannot be absorbed into a re-definition of $V(\phi)$. I presume the point is that later fluctuations of $\psi_i$ are considered, but this should be clarified. 6. The sentence below (B.17) does not appear right - probably a typo.

---

## Editorial Decision

awaiting_resubmission